# Transcriptomic and Phenotypic Analysis of CRISPR/Cas9-Mediated *gluk2* Knockout in Zebrafish

**DOI:** 10.3390/genes13081441

**Published:** 2022-08-13

**Authors:** Qianqian Yan, Wenhao Li, Xiaoting Gong, Ruiqin Hu, Liangbiao Chen

**Affiliations:** Key Laboratory of Aquacultural Resources and Utilization, Ministry of Education, College of Fisheries and Life Sciences, Shanghai Ocean University, Shanghai 201306, China

**Keywords:** *gluk2*, knockout, RNA-seq, behavior analysis, cold stress

## Abstract

As a subtype of kainite receptors (KARs), GluK2 plays a role in the perception of cold in the periphery sensory neuron. However, the molecular mechanism for *gluk2* on the cold stress in fish has not been reported. In this article, real-time PCR assays showed that *gluk2* was highly expressed in the brain and eyes of adult zebrafish. To study the functions of *gluk2*, gene knockout was carried out using the CRISPR/Cas9 system. According to RNA-seq analysis, we selected the differentially expressed genes (DEGs) that had significant differences in at least three tissues of the liver, gill, intestine, skin, brain, and eyes. Gene Ontology (GO) enrichment analysis revealed that *cry1ba*, *cry2*, *per1b*, *per2*, *hsp70.1*, *hsp70.2*, *hsp70l*, *hsp90aa1.1*, *hsp90aa1.2*, *hspb1*, *trpv1*, *slc27a1b*, *park2*, *ucp3*, and *METRNL* were significantly enriched in the ‘Response to temperature stimulus’ pathway. Through behavioral phenotyping assay, the *gluk2*^−/−^ larval mutant displayed obvious deficiency in cold stress. Furthermore, TUNEL (TdT-mediated dUTP Nick-End Labeling) staining proved that the gill apoptosis of *gluk2*^−/−^ mutant was increased approximately 60 times compared with the wild-type after gradient cooling to 8 °C for 15 h. Overall, our data suggested that *gluk2* was necessary for cold tolerance in zebrafish.

## 1. Introduction

As poikilothermic vertebrates, the behavior and thermoregulation of fish strongly depend on water and the environment [1]. When noxious heat or cold occurs, fish rely on the fast activation of the sensorimotor pathway and the exquisite thermoreceptors to avoid biochemical and physiological damages. The thermoregulatory center of fish is in the brain (preoptic area of the anterior hypothalamus) [2]. Caudal neurosecretory Dahlgren cells also play important roles in the response to ambient temperature and transient osmotic stress [3]. Besides, fish rely on the thermal receptors in trigeminal (TG) and dorsal root ganglia (DRG) neurons, which innervate the facial skin and the skin of the trunk and tail, respectively [1]. Temperature sensation was transmitted via the afferent sensory neurons that primarily emit small-diameter C-fibers and medium-diameter Aδ-fibers, which could be activated by distinct temperature thresholds [4]. At present, eleven kinds of temperature-sensitive transient receptor potential (thermoTRP) ion channels ranging from extreme cold to noxious heat are found in mammals [5]. In the absence of TRPV1 or TRPM3 ion channels, warm-excited-sensitive polymodal C-fibers were blunted. TRPM8 was predominantly expressed in cool-sensitive C-fibers, which is essential for both cold and warm perception [6]. Furthermore, TRPV2, TRPV3, and TRPM8 were lost in the evolution of zebrafish, which was associated with the adaptation of habitats and the survival of organisms [7].

Glutamate receptors are divided into ionotropic glutamate receptors (iGluRs) and metabotropic glutamate receptors (mGluRs) [8,9,10]. Kainate receptors (KARs) constitute a special subfamily of iGluRs that are made up of five subunits (GluK1 to GluK5). The excitatory ionotropic glutamate receptors are widely distributed and are particularly abundant in both presynaptic and postsynaptic sites of neuron membranes to regulate glutamate release [11]. KARs were located on sensory afferent fibers and DRG cells. A kainate-type glutamate receptor GluK2 was broadly expressed across GABAergic neurons and contributed to the development of synaptic plasticity [12]. Due to glutamate receptors’ ability to transmit chemical synaptic signals in the central nervous system, it validated that mGluR ligands had therapeutic utility in neurological and psychiatric disorders such as epilepsy, schizophrenia, and autism [13]. In mice, when lacking glutamate receptor 5 (GluR5), the responses to capsaicin or inflammatory pain would be significantly reduced [14]. It is known that TRPM8 with an activation threshold at ~26 °C, TRPA1 with an activation threshold < 17 °C, and TRPC5 with an activation threshold at 37–25 °C are all recognized as cold-sensitive channels, although some of their functions remain elusive [15]. Notably, GluK2 from zebrafish plays a role as a cold sensor, and the activation threshold is ≤18 °C [16]. GluK2 is expressed in membrane coupling, which interact with G proteins through the CTD domain [17]. When it functions as a metabotropic glutamate receptor, GluK2 transmits cold signals via Gi/o-coupled signaling [16]. What is more, it has been reported that Connexin 36 (Cx36) in the TG mediates orofacial pain hypersensitivity through GluK2 and TRPA1 signaling. However, different from the fact that TRPA1 activated mechanical allodynia, GluK2 only participates in cold allodynia as a temperature sensor, rather than both a mechanical and cold sensor [18].

However, currently, no study has focused on the kainite receptor in fish to cold stress. Zebrafish is a powerful vertebrate model organism on account of its highly genetic conservation to human and amenability to genome editing [19]. Here, we firstly use CRISPR/Cas9-mediated *gluk2* knockout in zebrafish and provide evidence about the biological function of *gluk2* through transcriptome sequencing analysis. Furthermore, we study the behavioral phenotyping of *gluk2*^−/−^ larvae in low temperatures, and the gills’ apoptosis of *gluk2*^−/−^ adult zebrafish through cold stress. Collectively, our results validate the importance of *gluk2* in cold adaptation, which provides a novel theoretical basis for improving temperature stress resistance in fish.

## 2. Materials and Methods

### 2.1. Animals and Embryos

Wild-type AB strain zebrafish were raised at 28 °C in a re-circulating system. Zebrafish embryos fertilized in vitro and raised under standard laboratory conditions. All mutants for experiment were zygotic embryos or adults.

### 2.2. Quantitative Real-Time PCR

The tissue samples included the liver, gill, blood, kidney, skin, intestine, brain, eyes, and eggs. Total RNA was extracted with the TRIzol reagent (Invitrogen). In this case, the internal control gene used for these analyses was the housekeeping gene 18S rRNA. All primer sequences are listed in Appendix A. Real-time PCR was carried out on a CFX96™ Real-Time PCR Detection System (Bio-Rad) using SYBR Premix Ex Taq™ II (Takara). The qPCR analysis was performed, and PCR conditions were as follows: 94 °C 3 min, followed by 30 cycles of 94 °C 20 s, 60 °C 20 s, and 72 °C 20 s. The expression level of the *gluk2* gene was calculated using the comparative threshold cycle (Ct) 2^−ΔΔCT^ method, and all samples were tested in triplicate. Relative mRNA levels were expressed as means ± SEM.

### 2.3. Gene Knockout

The gene of *gluk2* (ENSDARG00000113771) was disrupted using the CRISPR/Cas9 system. The exon sequences of the target gene were obtained through the Ensembl website (http://asia.ensembl.org/Danio_rerio/Gene/Summary?g=ENSDARG00000113771;r=16:1802307-2105556) (accessed on 2 June 2020) The target site of the *gluk2* gene was designed according to the ZiFiT website (http://zifit.partners.org/ZiFiT/Disclaimer.aspx) (accessed on 2 June 2020) and listed in Appendix A. The MAXIscriptTMT7 in Vitro Transcription Kit (AM1314) was purchased from Ambion (USA) and the GenCrispr NLS-Cas-NLS Nuclease (Z03389-100) was purchased from GenScript (China). The microinjection of zebrafish embryos was performed at the one-cell stage with 200 ng/µL gRNA and 800 ng/µL Cas9 protein mixture. The sequencing peak maps were performed using Chromas MFC Application software. The three-dimensional structures of GluK2 were predicted using the SWISS-MODEL Server (https://swissmodel.expasy.org/interactive).

### 2.4. Transcriptome Sequencing and RNA-Seq Data Processing

Total RNA was extracted using TRIzol reagent following the manufacturer’s protocol (Invitrogen). Quality and quantity measurements of the extracted RNA were determined using NanoDrop (Thermo Fisher Scientific). All samples were sent to the company (ANOOAD, Beijing, China) for RNA-seq. For each sample, at least 40 million 150-bp paired-end reads were generated by RNA-seq. Reads were mapped to the zebrafish reference genome (GRCz11, ENSEMBL annotation release 104, http://ftp.ensembl.org/pub/release-104/fasta/danio_rerio/dna/). The annotation and gtf files were downloaded from the Ensembl database (http://ftp.ensembl.org/pub/release-104/gtf/danio_rerio/). Hisat2-2.0.4 was used to map the reads to the reference genomes. SAMTools was used to first sort the bam files of the aligned reads by read name. The read counts were calculated using the featureCounts program. Then, the read counts of each sample were imported into the ‘DESeq2’ package for a differential expression analysis. The shrinkage of the logarithmic fold change (LFC) was calculated using empirical Bayes techniques integrated into DEseq2. These shrunken LFCs and their standard errors are used in the Wald tests for differential expression. The DEGs between the groups (GK2 vs. WT) wer identified with the criteria of the false discovery rate (FDR) adjusted *p*-value < 0.05 and | log2 (Fold Change) | > 1. GO terms and KEGG (Kyoto Encyclopedia of Genes and Genomes) pathway enrichment analyses with corrected *p*-value less than 0.05 were considered as significantly enriched. All statistical calculations were implemented by the ‘clusterProfiler’ R package. The clean read count was used to calculate the FPKM (expected number of fragments per kilobase of transcript sequence per millions base pairs sequenced), which was used to characterize the gene transcription abundance. The possible potential interactions between coding proteins were predicted, and the protein–protein interaction network was constructed for DEGs through the STRING database (https://string-db.org/) (accessed on 2 June 2020). After obtaining the PPI relationship, a network diagram was constructed using Cytoscape (version 3.9.0) combined with the CytoHubba plugin.

### 2.5. Behavior Analysis

Larval zebrafish at 7 days post fertilization (dpf) were placed individually into wells on a round 96-well plate (0.35 cm^2^ bottom area, 400 μL of circulating water for fish culture). Then, the 96-well plate was placed in Daniovision (Noldus, Wageningen, The Netherlands). The behavior of zebrafish in each well was monitored by Daniovision with a resolution of 1024 × 768 pixels at 25 frames per seconds (fps). The recorded video images were subjected to Ethovision XT11 (Noldus) to measure the behavior of zebrafish in each well. The distance moved within 10 min, time spent in the well center during 10 min, acceleration frequency, manic degree, rotation frequency, and active frequency were determined as parameters.

### 2.6. TUNEL Staining

Paraffin sections were prepared for immunofluorescence staining. Nuclear fragmentation was evaluated using the FITC Apoptosis Detection Kit (Vazyme, A11103) according to the manufacturer’s protocol. Subsequently, stained sections were counterstained with 500 ng/mL diamidine phenyl indoles (DAPI) for 10 min and observed with a fluorescence microscope (Zeiss). An average of the apoptotic cell proportion was calculated in 3 randomly acquired imaging fields (×10/×40) in each section. Fluorescence intensities were measured using ImageJ software.

### 2.7. Statistical Analysis

Data were analyzed using GraphPad Prism 8.0.1 (GraphPad, San Diego, CA, USA). All data were presented as mean ± SEM. Statistical significance between two groups was determined by the unpaired Student’s t-test and the Chi-Square Calculator (https://www.shuxuele.com/data/chi-square-calculator.html) (accessed on 2 June 2020). *p* < 0.05 was considered statistically significant, *p* < 0.01 was considered highly statistically significant, whereas *p* > 0.05 was considered to be nonsignificant.

## 3. Results

### 3.1. Distribution and Knockout of the gluk2 Gene in Zebrafish

We investigated the tissue expressions of *gluk2* in adult zebrafish, which were determined by qPCR in eleven types of tissue: Liver, gill, spleen, kidney, heart, muscle, intestine, skin, brain, eyes, and eggs. As shown in Figure 1A, *gluk2* was prominently expressed in the brain, followed by the eyes and eggs. The mutant allele of zebrafish *gluk2* was generated using CRISPR/Cas9. *gluk2*^*−*^^26*bp*^ contains a 26-base pair (bp) deletion, introducing a frameshift positioned to disrupt GluK2’s translation (Figure 1B). Compared to WT, the *gluk2* could not be detected from th mRNA level in the mutant (Figure 1C). P151L mutation can disrupt the cold sensitivity of GluK2 in CHO cells [16], and this lesion is predicted to generate a P151H mutation. Therefore, the N-terminal ATD of GluK2 required for cold-sensitivity was disrupted (Figure 1D). 

### 3.2. Functional Enrichment of DEGs

From GK2 to WT (GK2 vs. WT), in the liver, 1720 transcripts were identified as differentially expressed, among which 1180 were up-regulated and 540 were down-regulated. In intestine, 1851 transcripts were identified as differentially expressed, of which 992 were up-regulated and 859 were down-regulated. In the gills, 1573 transcripts were identified as differentially expressed, of which 753 were up-regulated and 820 were down-regulated. In the skin, 4330 transcripts were identified as differentially expressed, of which 1964 were up-regulated and 2366 were down-regulated. In the brain, 1272 transcripts were identified as differentially expressed, of which 593 were up-regulated and 679 were down-regulated. In the eyes, 2076 transcripts were identified as differentially expressed, of which 1060 were up-regulated and 1016 were down-regulated. Although *gluk2* was widely distributed in nerve cells in the brain, we obtained the largest number of differential genes in the skin. Thus, *gluk2* not only affected the central nervous system but also affected the expression of DEGs in the peripheral nervous system (Figure 2A).

To reveal the potential transcriptional regulation underlying the mutation of *gluk2* in various tissues, GO enrichment analyses were carried out with the differentially expressed genes in the pairwise comparison of groups (GK2 vs. WT) including the liver, gill, intestine, skin, brainm and eyes. To further investigate the function of the network, the identified DEGs were imported into the clusterProfiler for GO functional enrichment analysis. For different tissues, GO enrichment analyses of DEGs were primarily enriched in ‘Cellular modified amino acid metabolic process’, ‘Immune response’, ‘Peptidyl-amino acid modification’, ‘Response to radiation’, ‘ATP metabolic process’, ‘Sensory perception of light stimulus’, and ‘Nervous system process’ (Appendix A). The selected DEGs that all had significant differences in at least three of the six tissues were primarily enriched in ‘Immune response’, ‘Muscle contraction’, ‘piRNA metabolic process’, ‘Regulation of circadian rhythm’, and ‘Exogenous drug catabolic process’ pathways (Figure 2B). Furthermore, the identified circadian rhythm (*cry1ba*, *cry2*, *per1b*, *per2*), heat shock protein family (*hsp70.1*, *hsp70.2*, *hsp70l*, *hsp90aa1.1*, *hsp90aa1.2*, *hspb1*), and other temperature-related proteins (*trpv1*, *slc27a1b*, *park2*, *ucp3*, *METRNL*) were significantly enriched in the ‘Response to temperature stimulus’ pathway (Figure 2C). Down-regulation of heat shock protein expressions was mostly remarkable in the gill, intestine, brain, and eyes of the *gluk2*^−/−^ mutant, and *p*-values were calculated by DEseq2 after normalization. *per1b* and *cry1* are members of the period family genes. *per1b* is expressed in a circadian pattern in the suprachiasmatic nucleus. *per1b*, *per2*, *cry1*, and *cry2* all had significant differences in eyes that might affect visual function. However, the expression of period family genes in the skin showed an opposite trend compared with other tissues. This might result from the fact that skin cells had peripheral clocks that could function autonomously [20]. Except for the eyes, the expression of heat shock proteins (*hsp70.1*, *hsp70.2*, *hsp70l*) were significantly down-regulated in the gills and intestine, and *hsp90aa1.2* was also significantly down-regulated in the gills, intestine, and brain. The small heat shock protein (*hspb1*) was down-regulated in the liver, skin, and eyes. We speculated that the knockout of *gluk2* affected the stress response for zebrafish under normal growth conditions. In addition, *gluk2* might have other biological function in the eyes. Moreover, the knockout of *gluk2* up-regulated the expression of the heat-induced sensor (*trpv1*) in the skin and eyes. While *slc27a1b* was related to the PPAR-alpha pathway, which was significantly up-regulated in various tissues of the mutant. *park2* was a component of a multiprotein E3 ubiquitin ligase complex, that was related to Mitophagy. *slc27a1b* and *park2* were two key genes related to ‘Response to temperature stimulus’ pathway. The difference in expression of mitochondrial uncoupling proteins (*ucp3*) suggested that *gluk2* also affected transporter activity and oxidative phosphorylation uncoupler activity pathways (Figure 2D). 

### 3.3. Behavioral Profiling of Cold Stress in Zebrafish Larvae

Larval zebrafish tend to change their swim speed, swim vigor, and turning frequency the farther they are from their preferred temperature [1]. In a 96-well format, through the behavioral phenotyping analysis of 7 dpf larval zebrafish at 28 °C or 18 °C for 10 min, we could directly see the change in fish’s trajectory in their corresponding wells after cooling (Figure 3A,B). In the locomotion assay, the average active frequency of *gluk2*^−/−^ larval fish at 28 °C was even slightly higher than that of wild-type (Figure 3C–H), and we could not rule out the differences among individuals, and whether the deletion of *gluk2* could result in hyperactivity or epilepsy in the zebrafish central nervous system [21,22]. However, it was worth noting that, compared with 28 °C, the distance moved, time spent in center, acceleration frequency, manic degree, rotation frequency, and active frequency of the *gluk2*^−/−^ mutant at 18 °C were significantly higher than that of the wild-type. It seemed that the *gluk2*^−/−^ mutant could not enact normal behavioral strategies to cope with cold stress. 

### 3.4. Decreased Cold Tolerance of gluk2^−/−^ Mutant

Adult wild-type and *gluk2*^−/−^ mutants were reared at 28 °C for at least three generations. The fish were exposed to the same continuous, linear cooling scheme with water temperature declining from 28 °C to 18 °C and then declined to 8 °C (Figure 4A) [23]. After being exposed to 8 °C for 15 h, to compare their lower temperature limit (LTL), we found that the mutant began to lose body equilibrium and the survival rate was approximately 10–20% lower than that of the wild-type for subsequent 0.5-h intervals in a 5-h test. At 1.5 h the wild-type and *gluk2*^−/−^ mutant had a significant difference in the survival rate (Figure 4B). Since all fish died within 5 h, to evaluate cold-induced damage, TUNEL staining showed that the average apoptosis signal in the mutant’s gills was approximately 60 times higher than that of the wild-type within 5 h (Figure 4B–D). From the overall level, the brain was not as sensitive as the gills to cold. However, we still found the apoptosis signal in the genotype, which was more serious than that in wild-type (Appendix A). In summary, the disruption of *gluk2* could result in the impairment and the excessive apoptosis of tissues in cold stress. 

## 4. Discussion

In ectotherms, fish are sensitive to their aquatic environment [24]. An acute decrease in temperature affects the physiology, metabolism, and reproduction of fish that will result in huge economic losses in aquaculture [25,26,27,28]. Fish can employ navigational strategies to reach their preferred water temperature and avoid harmful conditions [1]. In the process, they sense environmental temperature, and ultimately transform this sensory information in the brain into behavioral strategies. In this article, the mutation of *gluk2* in zebrafish exhibited a partial defect in cold sensation, which changed the behavioral responses of larval zebrafish in low temperatures. Besides, the gills are a cold-sensitive tissue for fish in respiratory responses [29]. The excessive apoptosis in gills reduced the *gluk2*^−/−^ mutant’s resistance to cold-temperature exposure. Moreover, the carbohydrates and lipids of fish would be mobilized strategically after the energy homeostasis is disrupted under cold stress [30].

By RNA-Seq analysis, ‘Immune response’, ‘Muscle contraction’, ‘ATP metabolic process’ and ‘piRNA metabolic process’ pathways were most significantly enriched in DEGs between *gluk2*^−/−^ and the wild-type. Notably, when lacking the function to receive the signal of peripheral temperature, the ability of fish to resist the invasion of external pathogens would be greatly weakened [31,32]. It has been reported that TRPV1 can be activated in response to high-temperature stress, and the mRNA abundance of *trpv1* is correlated with the expression of pro-inflammatory cytokines in the Pre-Optic Area (POA) and the cytokine release in plasma [33]. At the same time, GluK2 sensing cold is a dynamin-dependent process regulated by releasing the peak Ca^2+^ transient [16]. Inevitably, the impairment of taking up Ca^2+^ would do great damage to muscle and cellular function [34].

*gluk2* was also significantly involved in ‘Regulation of circadian rhythm’, ‘Sensory perception of light stimulus’, ‘Exogenous drug catabolic process’, and ‘Nervous system process’ pathways. Furthermore, light and temperature represent two key environmental timing cues (zeitgebers), where the circadian clock enables animals to optimize their physiology and behavior in the brain that anticipates day/night cycles [35]. In this paper, the DEGs involved in the ‘Response to temperature stimulus’ pathway mostly include *cry1ba*, *cry2*, *per1b*, *per2*, *hsp70.1*, *hsp70.2*, *hsp70l*, *hsp90aa1.1*, *hsp90aa1.2*, and *park2*, while the cycling of the PER protein resulted from the cycling of period mRNA through self-sustained translation feedback loops [36]. This clockwork is highly complex and includes many additional collective components (PER, CRY, CLOCK, and BMAL1) that contribute to the periodicity of circadian molecular [37]. Studies have indicated that the disorder of the endogenous circadian clock is associated with neurodegenerative diseases, metabolic disorders, and inflammation [38,39]. In addition, when a thermal stressor is present, the transcription of HSPs can be activated and rapidly translated into proteins that act as molecular chaperones to protect the cell against denatured proteins [40]. Our study demonstrated that the mutation of *gluk2* caused the disorder of the circadian rhythm. Besides, the expression of HSPs and TRPs was mainly synchronized to the light/dark (LD) cycle [41]. However, studies also showed that the methylation modification pattern of a single arginine methylation site on heat shock protein 68 (HSP68) could inhibit the expression of PER [39]. Therefore, whether HSP is involved in gene transcription to regulate the circadian rhythm by epigenetic modification or circadian rhythm disruption leading to the decreased expression of heat shock protein remains unclear [35,42,43].

Research showed that fish had less physiological and ecological susceptibility to cold temperatures compared with high temperatures. For example, environmental temperature could influence the basal heat shock protein mRNA levels and the heat shock response (HSR) in lake whitefish (*Coregonus clupeaformis*). Under repeated thermal stress, it could result in a downregulation of inducible *hsps* throughout embryogenesis [44]. For cold-adapted lake trout (*Salvelinus namaycush*), the plasticity of cardiac HSP70 and HSP90 was beneficial for them to cope with thermal stress [45]. Taken together, we believe that this functional annotation provided from the classification of transcripts from existing protein databases would provide a basis for further work in genetic or genomic studies on temperature sensing of *gluk2*.

## 5. Conclusions

In this study, the knockout of *gluk2* reduced the cold tolerance in zebrafish. Additionally, it was accompanied with a change in the behavioral phenotype for the *gluk2*^−/−^ larval mutant in cold stress. The temperature-related DEGs were mostly associated with the circadian rhythm and the heat shock protein. In brief, this study provides fundamental insight into the molecular mechanism that *gluk2* is essential for fish to deal with cold stress.

## Figures and Tables

**Figure 1 genes-13-01441-f001:**
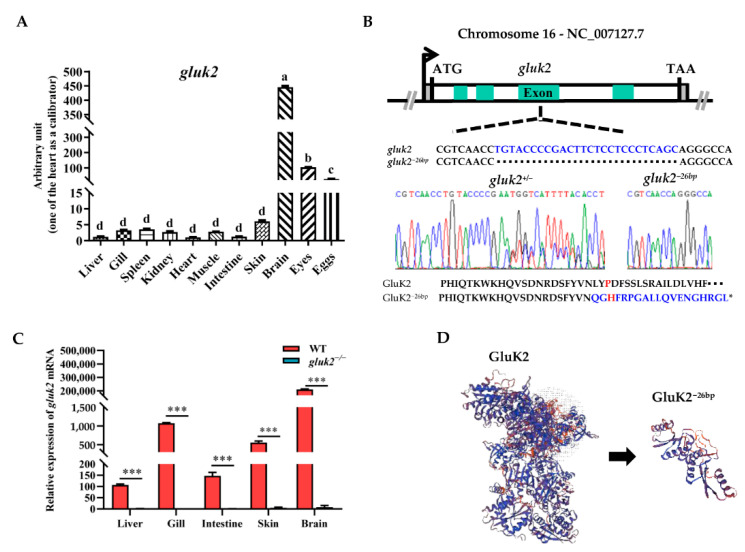
Distribution and knockout of the *gluk2* gene in zebrafish. (**A**). The relative mRNA levels of *gluk2* are determined in selected tissues of wild-type. Analysis of variance (ANOVA) and the Duncan test are applied on the data, *p* < 2 × 10^−16^. a−d: The same letter indicates no difference between means as determined (*p* < 0.05), different letters indicate a significant difference (*p* < 0.05). (**B**). *gluk2* locus, with *gluk2^−^*^26bp^ mutation and a premature stop in *gluk2^−^*^26bp^. Asterisk indicates stop codon. (**C**). The relative mRNA levels of *gluk2* are determined in wild-type and *gluk2*^−/−^ mutant. Values plotted are means ± SEM, *** *p* < 0.001, compared with control. (**D**). The predicted three-dimensional structures of GluK2 and GluK2^−26bp^.

**Figure 2 genes-13-01441-f002:**
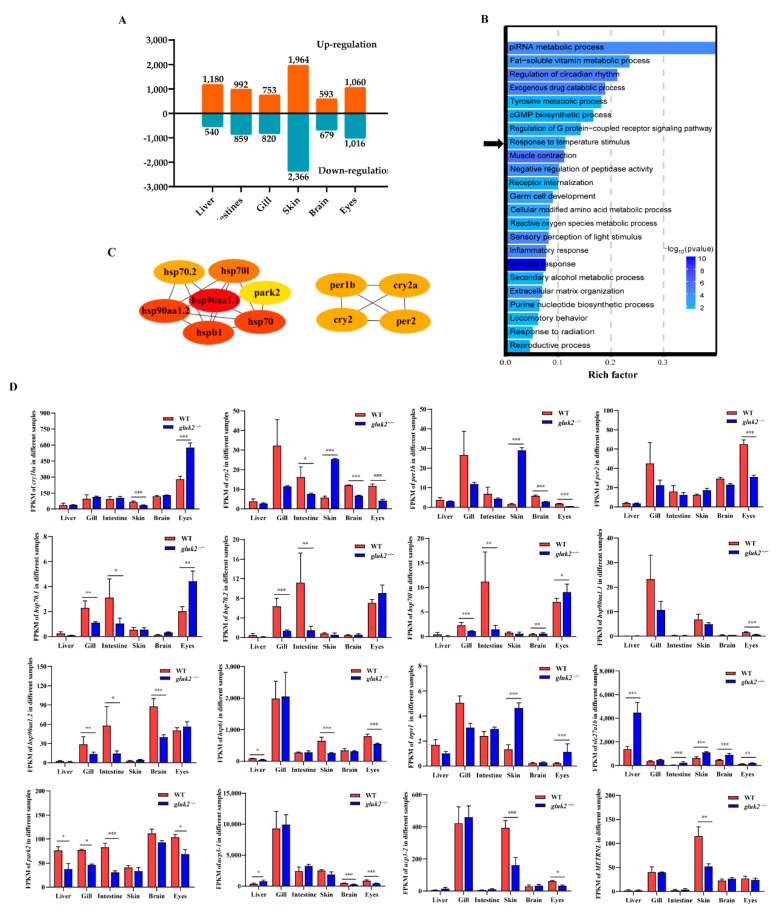
Divergent pathway and gene expression between wild-type and *gluk2*^−/−^ mutant. (**A**) Number of DEGs obtained from the six tissues at the wild-type and *gluk2*^−/−^ mutant. (**B**) Gene Ontology (GO) enrichment analysis of differentially expressed genes identified from the wild-type and *gluk2*^−/−^ mutant. The abscissa represents the GO terms, and the ordinate represents the number of target genes. Black arrow indicates the ‘Response to temperature stimulus’ pathway. (**C**) Protein–protein interaction network for *gluk2*^−/−^ expressed in the response to temperature stimulus pathway. (**D**) Fragments Per Kilobase per Million (FPKM) of divergent gene expression related to temperature signaling pathway in different tissues. Values plotted are means ± SEM, * *p* < 0.05, ** *p* < 0.01, *** *p* < 0.001, compared with control.

**Figure 3 genes-13-01441-f003:**
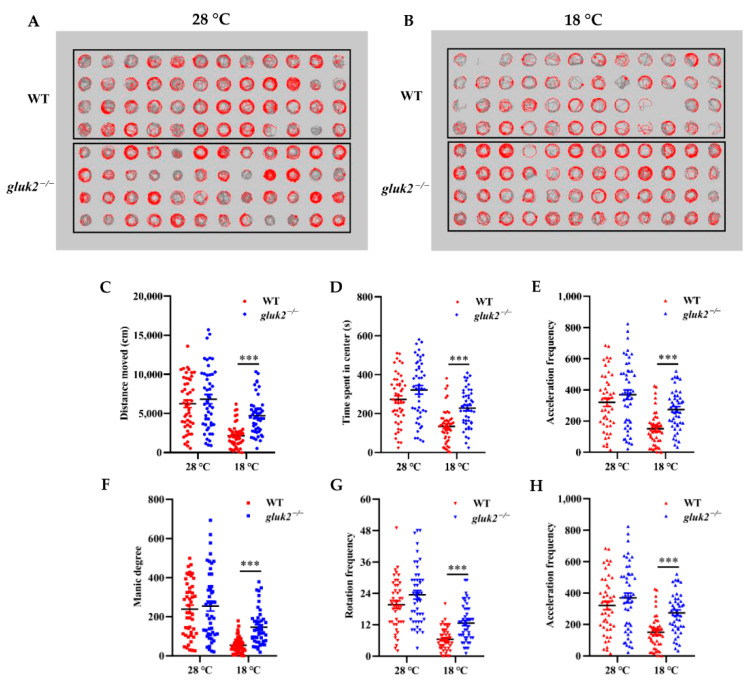
Effects of low temperature on the 7 dpf wild-type and *gluk2*^−/−^ zebrafish locomotor activities. (**A**) Wild-type and *gluk2*^−/−^ zebrafish in a 96-well plate during tracking at 28 °C. (**B**) Wild-type and *gluk2*^−/−^ zebrafish in a 96-well plate during tracking at 18 °C. (**C**) Distance moved by the wild-type and *gluk2*^−/−^ zebrafish larvae at 28 °C and 18 °C. (**D**) Time spent in center by the wild-type and *gluk2*^−/−^ zebrafish larvae at 28 °C and 18 °C. (**E**) Acceleration frequency by the wild-type and *gluk2*^−/−^ zebrafish larvae at 28 °C and 18 °C. (**F**) Manic degree by the wild-type and *gluk2*^−/−^ zebrafish larvae at 28 °C and 18 °C. (**G**) Rotation frequency by the wild-type and *gluk2*^−/−^ zebrafish larvae at 28 °C and 18 °C. (**H**) Active frequency by the wild-type and *gluk2*^−/−^ zebrafish larvae at 28 °C and 18 °C. Values plotted are means ± SEM, *** *p* < 0.001, compared with control.

**Figure 4 genes-13-01441-f004:**
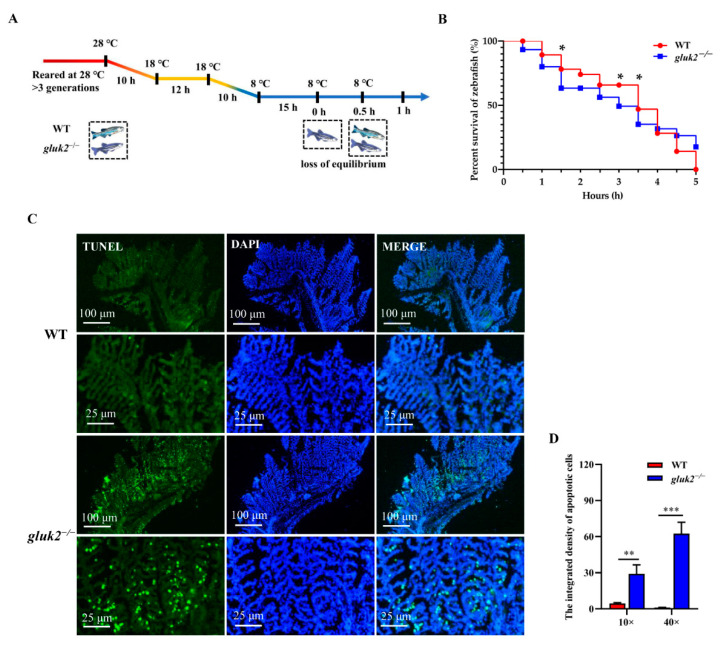
Distinct cold tolerance in wild-type and *gluk2*^−/−^ zebrafish under the cooling scheme. (**A**) The changes in the body equilibrium of wild-type and *gluk2*^−/−^ zebrafish over time at 8 °C. (**B**) The percent survival of wild-type and *gluk2*^−/−^ zebrafish at 8 °C from 0 h to 5 h. (**C**) TUNEL assays in the gills of wild-type and genotypes, indicating a more serious apoptosis in *gluk2*^−/−^ mutant than in wild-type at the same cold temperature. Scale bar are 100 μm and 25 μm. (**D**) Comparison of apoptotic cell populations in the gills of wild-type and *gluk2*^−/−^ zebrafish over time at 8 °C. Values plotted are means ± SEM, * *p* < 0.05, ** *p* < 0.01, *** *p* < 0.001, compared with control, and are based on at least three biological replicates, with each replicate having at least 3 individuals at each time point.

## Data Availability

The RNA-seq data have been submitted to the National Center for Biotechnology Information Sequence ReadArchive (http://www.ncbi.nlm.nih.gov/sra, accessed on 6 June 2026) under BioProject accession numbers PRJNA847767.

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
