# Peer review of "Transcriptomic and Phenotypic Analysis of CRISPR/Cas9-Mediated *gluk2* Knockout in Zebrafish"

_genes, 2022, doi:10.3390/genes13081441_

Round 1

Reviewer 1 Report

In this manuscript by Yan et al., the authors describe a role for gluk2, a subtype of kainite receptors, in zebrafish cold tolerance. The authors use the zebrafish model to generate a gluk2 Knockout model and analyse the transcriptomic and phenotypic outcomes. This work is interesting as it validates the role of Gluk2 in zebrafish cold tolerance and it unravels the molecular blueprint related to Gluk2 activity in the different organs.

However, to this reviewer, there are still some important issues that need to be addressed and overall writing could be significantly improved:

1-     The in situ experiment shows a ubiquitous expression of gluk2 at different stages of development, however the images are of poor quality and there is no control sense probe. It would be better to download some better quality images and to provide a sense probe labeling just to be sure that the labeling reflects gluk2 expression.

2-     The mutant generated in this paper is well described, however not validated. It is important for the authors to validate this mutant by looking at the levels of Gluk2 protein. It is sometimes difficult to have working antibodies in zebrafish but I guess there is a commercially available one (Lavinia sheets, Excessive activation of ionotropic glutamate receptors induces apoptotic hair-cell death independent of afferent and efferent innervation, Scientific reports, 2017). If not, authors might need to rescue, at least one of the phenotypes by injecting gluk2 mRNA or look for gluk2 mRNA expression in controls vs mutants for nonsense mRNA decay following mutation.

3-     It is not clear if authors are using zygotic embryos/adults or Maternal Zygotic ones?

4-     Figure 4B: Authors might need to show at what time the two groups show a significant difference.

5-     The stats are not clear, it would be better if the statistical test is highlighted for each experiment as well as the number of embryos. Statistical analysis in materials and methods might need to be more elaborate (normality test..)

6-     The chief experiment in this paper is the transcriptomic analysis. However, authors need to discuss these results to link the molecular blueprint of Gluk2 activity to cold tolerance. Discussion might need to revisited, some parts are not clear.

7-     Figure 2D: Divergent gene expression vary differently in different tissues, any comment on that, e.g. skin vs brain and eyes for per1b. This might need to be at least discussed.

8-     The English might need major revision

9-     Authors might need to reconsider all nomenclatures of genes vs proteins in this paper by following zebrafish nomenclature guidelines.

10-   Figure S2: the lack of significant difference between WT and Gluk2-/- groups at 10x is quite intriguing!

Reviewer 2 Report

The manuscript by Yan et al. studies the function of GluK2 gene in zebrafish by CRISPR knockout. After comparative transcriptome analysis in different tissues using RNA-seq, they find “response to temperature stimulus” as one of the functions that are affected by GluK2 knockout. Combining the RNAseq results with subsequent behavior and cold temperature tolerance tests, they conclude that GluK2 is required for zebrafish to tolerate cold temperatures. However, due to poor experimental design and data analysis, their conclusion is not convincing.

Major points:

1.     It is crucial for the authors to show how they validated the CRISPR knockout. For this, they need to show both the genotyping results and the examination of protein expression. In line 171-172, it says “… is predicted to generate a non-functional receptor”. Since a successful knockout is the basis for the whole paper, they must confirm the WT protein is successfully depleted after the knockout and should not rely on predictions.

2.     They need to show how the mutants look like.

3.     Figure 1B, are there statistical differences among the samples? What statistical tests were used?

4.     Since GluK2 shows the highest level of expression in brain (Figure 1B), one would expect it has stronger impact in this tissue than in other tissues. However, in Figure 2A, the number of DEGs is the smallest in brain. What is the authors’ explanation for this?

5.     For gene ontology analysis, what gene list was used? Why did the authors use common genes shared by at least three tissues not all the six tissues?

6.     Are the genes mentioned by the authors in line 218-220 top genes in the DEG lists? Or are they just arbitrarily chosen?

7.     “Response to temperature stimulus” is not the top GO term, why do the authors think this is the main function affected by GluK2 KO?

8.     Figure 2D, how were the p-values calculated? Were they calculated by Dseq2 after normalization or by using FPKM values?

9.     Figure 3 C & D, the authors need to explain in detail, in the Materials and Methods section, how the measurements were done.

10.  Figure 4B, no statistics was done.

11.  Figure 4C and D and Figure S2, the nuclei in these images are indistinguishable. How were the quantifications carried out?  Why do the percentages differ so much between 10X and 40X?

Minor points:

1.     Line 106, what aspects of the pictures were adjusted by Photoshop?

2.     Line 120, 5G of data is not meaningful, instead, the authors should use number of reads generated.

3.     Line 131, for DGE analysis, a fold change of 1.5 or 2 (equivalent to a log2 scale of 0.58 or 1) is more commonly used as the threshold. Why were the authors use |log2 (Fold Change) | >2 (=Fold change >4)? Is it because they got too many DE genes from their analysis?

4.     What statistical tests did they use for discovering DGEs in their RNAseq analysis?

5.     Figure 1 A, what is the difference between a, b, and c? Are they just showing three images of the same staining? The image quality (lighting, magnification, background, and resolution) needs to be improved.

6.     What is the purpose of Figure 3A&B? What information can we get from these images?

7.     The overall writing of the paper needs to be improved.

Author Response

  1. It is crucial for the authors to show how they validated the CRISPR knockout. For this, they need to show both the genotyping results and the examination of protein expression. In line 171-172, it says “… is predicted to generate a non-functional receptor”. Since a successful knockout is the basis for the whole paper, they must confirm the WT protein is successfully depleted after the knockout and should not rely on predictions.

Response: (line160-165). Thanks for your advice. I have tried the antibody of GluK2, it is regret that Western blot showed the antibody has poor specificity. Therefore, to further validate this mutant, I add the gluk2+/- and gluk2-/- mutants’ sequencing peak maps as evidence (Figure 1B). As is known in the paper (Gong et al. 2019), on the condition that the structure of GluK2 still remains integrity. From their paper, P151L single mutation can disrupt the cold sensitivity of GluK2 in CHO cells. In this paper, gluk2-26bp contains a 26-base pair (bp) deletion, introducing a frameshift positioned to generate a P151H mutation and a stop codon in advance. Therefore, we think that the function of GluK2 in mutant is disrupted. Furthermore, we design qPCR primer to detect the mutant from mRNA transcriptional level, compared to WT, the gluk2 cannot be detected by mRNA level.

  1. They need to show how the mutants look like.

Response: (Figure1B,C). Thanks for your advice. We have described the mutant again. Compared with WT, the gluk2-/- mutant looks normal. It has no significant in appearance. However, according to DNA sequencing and qPCR, we can examine this mutation.

  1. Figure 1B, are there statistical differences among the samples? What statistical tests were used?

Response: (line178-179). Thanks for your advice. We have modified the content. Analysis of variance (ANOVA) and the Duncan test were applied on the data. The significant differences are marked on Figure 1A. The P values for the ANOVO test are as follows: P <2e-16.

  1. Since GluK2 shows the highest level of expression in brain (Figure 1B), one would expect it has stronger impact in this tissue than in other tissues. However, in Figure 2A, the number of DEGs is the smallest in brain. What is the authors’ explanation for this?

Response: (line 193-196, line217-219). Thanks for your advice. We have modified the content. Although gluk2 is widely distributed in nerve cells in the brain, we obtain the largest number of differential genes in the skin. Thus, gluk2 not only affects the central nervous system, but also affect the expression of DEGs in the peripheral nervous system. And skin is an important organ for temperature perception, which will be an entry point for the research. The expression of Period family genes in skin showed an opposite trend compared with other tissues. This may result from that skin cells have peripheral clocks that can function autonomously.

Reference resources:

Matsui, Mary S et al. “Biological Rhythms in the Skin.” International journal of molecular sciences vol. 17,6 801. 24 May. 2016, doi:10.3390/ijms17060801.

Sporl F., Korge S., Jurchott K., Wunderskirchner M., Schellenberg K., Heins S., Specht A., Stoll C., Klemz R., Maier B., et al. Kruppel-like factor 9 is a circadian transcription factor in human epidermis that controls proliferation of keratinocytes. Proc. Natl. Acad. Sci. USA. 2012;109:10903–10908. doi: 10.1073/pnas.1118641109.

  1. For gene ontology analysis, what gene list was used? Why did the authors use common genes shared by at least three tissues not all the six tissues?

Response: Thanks for your advice. At the first step, we selected all DEGs (p-value < 0.05 and | log2 (Fold Change) | > 1) in six tissues as union. We have tried selecting genes that have significant difference in all the six tissues, but because of the different function of tissues, the DEGs are almost different. It will affect the effectiveness of gene ontology analysis. Therefore, we use common genes shared by at least three tissues. In fact, we also tried using at least two/three/four tissues of the six tissues, the results are similar. At last, we decided to use common genes shared by at least three tissues.

  1. Are the genes mentioned by the authors in line 218-220 top genes in the DEG lists? Or are they just arbitrarily chosen?

Response: (line 208-211).The genes mentioned are all genes related to ‘Response to temperature stimulus’ pathway in the DEG lists. We have made changes to make the order of appearance in the whole article and the picture is the same.

  1. “Response to temperature stimulus” is not the top GO term, why do the authors think this is the main function affected byGluK2 KO?

Response: (line 343-345). Thanks for your advice. Fish is a poikilothermic (cold-blooded). We believe that GluK2 from zebrafish is a cold sensor (Gong et al., 2019). When lack of gluk2, the ability for fish to perceive temperature will change. And yet temperature affects a wide range of biological processes such as immunity (Scharsack et al., 2022), reproduction and growth (Lema et al., 2022). Therefore, although “Response to temperature stimulus” is not the top GO term, we still believe that Go terms such as ‘Immune response’, ‘piRNA metabolic process’ , ‘Regulation of circadian rhythm’ and ‘Exogenous drug catabolic process’ pathways are resulted from lacking temperature sensing.

Reference resources:

(Scharsack, Jörn Peter, and Frederik Franke. “Temperature effects on teleost immunity in the light of climate change.” Journal of fish biology, 10.1111/jfb.15163. 14 Jul. 2022, doi:10.1111/jfb.15163)

(Lema, Sean C et al. “Accustomed to the heat: Temperature and thyroid hormone influences on oogenesis and gonadal steroidogenesis pathways vary among populations of Amargosa pupfish (Cyprinodon nevadensis amargosae).” Comparative biochemistry and physiology. Part A, Molecular & integrative physiology, vol. 272 111280. 25 Jul. 2022, doi:10.1016/j.cbpa.2022.111280)

  1. Figure 2D, how were the p-values calculated? Were they calculated by Dseq2 after normalization or by using FPKM values?

Response: (line 214). Thank you for your question. p-values were calculated by Dseq2 after normalization.

  1. Figure 3 C & D, the authors need to explain in detail, in the Materials and Methods section, how the measurements were done.

Response: (line138-140). Thank you for your advice. The measurements of Figure 3 C & D were calculated automatically when the larval zebrafish swimming (Figure3A,B). The parameters in Figure 3 C & D are similar to Figure 3E-H. I have explained in Materials and Methods section (line138-140).

  1. Figure 4B, no statistics was done.

Response: (line 285-286). Thanks for your advice. We have added the description that statistical significance between two groups was determined by the Chi-Square Calculator (https://www.shuxuele.com/data/chi-square-calculator.html). We have sign ‘star’ in Figure 4B and add the description in line 285-286. ‘At 1.5 h, they showed a significant difference.’

  1. Figure 4C and D and Figure S2, the nuclei in these images are indistinguishable. How were the quantifications carried out?  Why do the percentages differ so much between 10X and 40X?

   Response: (Figure 4C,D and Figure S2), (line 289-290). Thanks for your advice. We use TUNEL image to quantify the apoptotic cells. According to the unpaired Student’s t-test, statistical significance was determined. It has statistically significant with p < 0.05 in 10X and 40X. The significant differences are marked on Figure 4C,D and Figure S2 and we have improved the description of experimental results. ‘From the overall level, brain is not as sensitive as gill to cold. But we can still find the apoptosis signal in genotype is more serious than that in wild-type (Figure S2).’

Minor points:

  1. Line 106, what aspects of the pictures were adjusted by Photoshop?

Response: (Figure4C, S2). Only the pictures signed ‘MERGE’ were adjusted by Photoshop.

  1. Line 120, 5G of data is not meaningful, instead, the authors should use number of reads gen

Response: (line107-108). Thank you for your advice. I have deleted ‘at least 5 G of data were generated by RNA-seq’. According to calculating, there are at least 40 million 150-bp paired-end reads were generated by RNA-seq.

  1. Line 131, for DGE analysis, a fold change of 1.5 or 2 (equivalent to a log2 scale of 0.58 or 1) is more commonly used as the threshold. Why were the authors use |log2 (Fold Change) | >2 (=Fold change >4)? Is it because they got too many DE genes from their analysis?

Response: (line119-120). Thanks for your suggestion in the detail. This is my clerical error. I have corrected| log2 (Fold Change) | > 1’ is the real threshold for DEG analysis.

  1. What statistical tests did they use for discovering DGEs in their RNAseq analysis?

Response: (line115-117). I have described this part. The DEGs were calculated using empirical Bayes techniques integrated into DEseq2 to estimate priors for log fold change and dispersion.

  1. Figure 1 A, what is the difference between a, b, and c? Are they just showing three images of the same staining? The image quality (lighting, magnification, background, and resolution) needs to be improved.

Response: Thank you for your advice. Because of the poor design of in situ experiment, the result is not accurate. To ensure the authenticity of scientific research, I delete the images and the related description in the paper. In fact, the sequencing of the Zebrafish transcriptome from a range of tissues and developmental stages showed that gluk2 expressed in embryo 5 dpf (https://www.ncbi.nlm.nih.gov/gene/556013). Figure 1A also showed that gluk2 is expressed in adult eggs. Moreover, due to this part is not very close correlated with the other results of the paper. Therefore, I think that in situ experiment is not necessary for the whole experiment. I have decided to delete this part.

  1. What is the purpose of Figure 3A&B? What information can we get from these images?

Response: (line233-235). Considering that using the same 96-well plate of fishes could exclude the individual differences. From Figure3A&B, we can see the change of fishes’ trajectory in their corresponding wells after cooling (from 28℃ to 18℃). It is more intuitive to see their trajectories.

  1. The overall writing of the paper needs to be improved.

Response: Thank you for your advice. I have improved the overall writing of the paper.

Round 2

Reviewer 1 Report

The manuscript has been improved, but to me there are still some slight issues regarding the English, especially when it comes to tenses.

One minor point to be clarified: the letters d, a, b and c in Figure 1A.

Author Response

The manuscript has been improved, but to me there are still some slight issues regarding the English, especially when it comes to tenses.

Response: Thanks for your comments. I have improved my writing and pay attention to the manuscript’s tenses.

One minor point to be clarified: the letters d, a, b and c in Figure 1A

Response: (line 179-180). Thanks for your advice. Analysis of variance (ANOVA) and the Duncan test were applied on the data. The significant differences are marked on Figure 1A. The P values for the ANOVO test are as follows: P <2e-16.

Reviewer 2 Report

The revised manuscript has been clearly improved. I would suggest the authors address the following points:

1.     It is nice to see their genotyping result (Figure 1B), which shows a successful deletion of the 26 bp sequence in the gluk2 exon. It is also nice that the authors further confirm the depletion of WT mRNA using Q-PCR (Figure 1C), however, it is surprising that the variations among replicates in each group are so small, which can be seen from the length of the error bars. Even more surprising is Figure 1A, which almost shows no variations among the triplicates. How many fish were used in each replicate and why are the variations among triplicates so small?

2.     For TUNEL staining (Figure 4C and D and Figure S2), did the authors count cell numbers or calculate signal intensity? From their description, it seems they counted cell numbers and calculated cell proportion (FITC positive/DAPI positive). However, since it is impossible to identify single nuclei in their images, it is difficult to understand how they count the cells.

3.     For DGE analysis (line115-117), yes, Deseq2 employs empirical Bayes technique for shrinkage of LFC, what hypothesis testing did the authors do thereafter? Wald t-test?

4.     Figure 2D, some of the genes are downregulated while others are upregulated after gluk2 KO, the authors need to discuss this in discussion.
